# Everlasting impact of initial perturbations on first-passage times of non-Markovian random walks

N. Levernier[1], T. V. Mendes [2], O. Bénichou [3] ✉, R. Voituriez[3,4] & T. Guérin[2]

Persistence, defined as the probability that a signal has not reached a threshold up to a given observation time, plays a crucial role in the theory of random processes. Often, persistence decays algebraically with time with non trivial exponents. However, general analytical methods to calculate persistence exponents cannot be applied to the ubiquitous case of non-Markovian systems relaxing transiently after an imposed initial perturbation. Here, we introduce a theoretical framework that enables the non-perturbative determination of persistence exponents of Gaussian non-Markovian processes with non stationary dynamics relaxing to a steady state after an initial perturbation. Two situations are analyzed: either the system is subjected to a temperature quench at initial time, or its past trajectory is assumed to have been observed and thus known. Our theory covers the case of spatial dimension higher than one, opening the way to characterize non-trivial reaction kinetics for complex systems with non-equilibrium initial conditions.

The persistence $S(t)$ is the probability that a random process $x(t)$ has not reached a threshold up to time $t$[1,2]. This quantity is a natural tool in non equilibrium statistical physics to probe the history of various systems undergoing phase ordering[3–5] or reaction diffusion dynamics[2], or to quantify the efficiency of target search problems[6–15]. It has been recognized that the long time decay of persistence is often algebraic, $S(t) \sim t^{-\theta}$, where the persistence exponent $\theta$ is non trivial as soon as the process is non-Markovian (i.e., displays memory effects). This has triggered a number of experimental[16–20] and theoretical[4,5,21–29] studies. A striking example is given by the case of a spatially extended field that obeys a simple diffusion equation, which represents for example the height of a fluctuating interface or the local concentration of diffusive particles. The persistence exponents quantify in these examples the probability that the field at a given point has not reached a given threshold value at time $t$, starting from a random initial configuration. Despite the simplicity of the diffusion equation, it turns out that the determination of $\theta$ for this class of problems is a tour de force[21,22,28,30,31].

More generally, even for seemingly simple Gaussian dynamics where all correlation functions are known, $\theta$ is generally non-trivial and not known in closed form. In fact, the exponent $\theta$ depends on the full history of the process, and it is in general difficult to extract it from the correlation function[2]. This has triggered an intense theoretical activity for its determination. Existing approaches to quantify persistence exponents of Gaussian processes can be classified according to the nature, stationary or not, of the increments $x(t + \tau) - x(t)$. If these increments are stationary at all times, meaning that their statistics do not depend on the observation time $t$ (such as in the case of the fractional Brownian motion), $\theta$ is exactly known[28,29,32] (in $d$ dimensions, for scale-invariant processes with stationary increments[32], $\theta = 1 - Hd$). In the opposite case where the increments always depend on the observation time $t$ and thus never reach a stationary dynamics (i.e., are stationary at no times), persistence exponents have been calculated for the specific cases of the random acceleration process[33,34] or systems in which the dynamics

[1]Aix Marseille Univ., Université de Toulon, CNRS, CPT, Turing Center for Living Systems, 13009 Marseille, France. [2]Laboratoire Ondes et Matière d'Aquitaine, University of Bordeaux, Unité Mixte de Recherche 5798, CNRS, F-33400 Talence, France. [3]Laboratoire de Physique Théorique de la Matière Condensée, CNRS/UPMC, 4 Place Jussieu, 75005 Paris, France. [4]Laboratoire Jean Perrin, CNRS/UPMC, 4 Place Jussieu, 75005 Paris, France. ✉e-mail: benichou@lptmc.jussieu.fr

occurs at zero temperature[3–5,21,22,30,31,35,36], and is thus deterministic with random initial conditions. In the latter case, exact results for the persistence exponent are scarce[4,30,31], and the theoretical determination of $\theta$ generally relies on approximate methods.

However, numerous physical situations display a relaxation dynamics – typically after an initial perturbation – that becomes stationary only after a transient regime. This is the rule for processes interacting with many degrees of freedom, subjected to thermal fluctuations during the dynamics, but prepared in a non-equilibrium or perturbed state. As a prototypical example, consider a tagged monomer of a flexible polymer initially equilibrated at a temperature $T \neq 1$, and quenched to a different temperature $T_0 = 1$ at time $t \geq 0$. The dynamics of the tagged monomer keeps transiently track of this initial perturbation, and relaxes to the equilibrium state at $T_0$ with stationary increments. Persistence properties for such process with non stationary increments (ie displaying aging), which in this example are instrumental to quantify the reaction kinetics of the polymer with a given reactive site, remain largely unknown. In fact, there is a fundamental reason why standard methods to calculate persistence exponents cannot be applied for transiently aging processes (see SI, Sections A and B where we show that for transiently aging processes, the independent interval approximation, which is usually applied to calculate the statistics of zero crossing of the Gaussian process obtained after Lamperti transform, cannot be applied since intervals between zero crossing become ill-defined). The only available results for similar problems are limited to one-dimensional processes and provide bounds for the persistence exponents as well as perturbative expansions for weakly non-Markovian processes[28].

Here, we develop a general theoretical framework that enables the determination of the persistence exponents of general Gaussian processes displaying such transient aging dynamics. We stress that these Gaussian processes are non-Markovian (display memory effects), and appear in a wide range of contexts[37–46]. Our method enables us to reveal and quantify the impact of initial conditions, such as a temperature quench, on the persistence exponent. We also consider the case where the past trajectory of the stochastic process is known, e.g. because it has been observed. We show that the very observation of this past trajectory modifies the persistence exponent which is quantified by our approach. Importantly, our theory covers the physically relevant and widely unexplored case of persistence for non-Markovian random walkers living in a space of dimension higher than one.

## Results

We first consider a one-dimensional isotropic non-Markovian Gaussian stochastic process $x(t)$, which represents the position of a random walker at time $t$. It is entirely defined by its mean value, assumed for simplicity to be constant with time (unbiased process), and its covariance $\mathrm{Cov}(x(t), x(t')) = \sigma_0(t, t')$. This covariance is assumed to be given and to take the standard self-similar scaling[2] form at long times $t, t' \gg 1$, $\sigma_0(t, t') \sim t^{2H} G(t/t') \equiv \sigma(t, t')$, where $H$ is the usual Hurst exponent. We chose our units of time so that $G(1) = 1$. At long times, the mean square displacement $\sigma(t, t) = t^{2H}$ is assumed to diverge so that the particle does not remain close to its initial position, which leads to $H > 0$. Furthermore, we assume that the statistics of the increments $x(t + \tau) - x(t)$ become stationary at long times, i.e. become independent of the observation time $t$ when $t \to \infty$. This implies the existence of a transient regime associated to the progressive decay of the memory of the initial state, and defines a stationary covariance $\sigma_s$ given by

$$\sigma_s(\tau, \tau') = \lim_{t \to \infty} \langle [x(t + \tau) - x(t)][x(t + \tau') - x(t)] \rangle. \quad (1)$$

Of note, the persistence exponent $\theta$ is known to be given by $\theta = 1 - H$ under the stronger hypothesis that the statistics of the increments is stationary at any time (i.e., when $\sigma_s = \sigma_0$)[28,29]. The class of random walks that we consider here covers a broad spectrum of non-

Markovian processes used in physics, and in particular both subdiffusive ($H < 1/2$) and superdiffusive ($H > 1/2$) walks.

### Theoretical method to determine $\theta$

Our starting point to calculate the statistics of the first passage time (FPT) to the origin $x = 0$ is the following generalization of the renewal equation[6]

$$p(0, t) = \int_0^t d\tau f(\tau) p(0, t | \text{FPT} = \tau), \quad (2)$$

which results from a partition over the first-passage event. In this equation, $p(0, t)$ stands for the probability density that the random walker is at position $x = 0$ at time $t$, $f$ is the first-passage time density and $p(0, t | \text{FPT} = \tau)$ is the probability density that $x = 0$ at time $t$ given that the first-passage event occurred at time $\tau$.

To proceed further, we assume that the stochastic process in the future of the FPT, defined by $y(t) \equiv x(t + \text{FPT})$, is Gaussian with so far undetermined mean $\mu(t)$ and covariance $\sigma_\pi(t, t')$. Such Gaussian approximation has proved successful to seize memory effects to predict mean first-passage times of Gaussian random walkers in confinement with stationary increments[12,13,47]; in the present context simulations show the broad validity of this hypothesis (see SI, Fig. S1). A first result of our approach is that the exponent $\theta$ is linked to the large time behavior of $\sigma_\pi(t, t)$, which is found from Eq. (2) to behave like (see SI, Section D)

$$\sigma_\pi(t, t) - \sigma_0(t, t) \underset{t \to \infty}{\propto} t^{2H - \theta}. \quad (3)$$

This means that the calculation of the exponent $\theta$ amounts to that of the covariance $\sigma_\pi(t, t')$ of the trajectories in the late future of the first-passage.

Relying on a generalization of Eq. (2) to link the two-time joint probability distribution functions of $x(t_1), x(t_2)$ and the FPT density, we obtain a self-consistent equation for the distribution of trajectories in the future of the FPT, leading in the large time limit to (see SI, Section D for details):

$$\int_0^\infty \frac{dt}{t^H} \left\{ \rho(t + \tau, t + \tau') - \rho(t + \tau, t) \frac{\sigma(t + \tau', t)}{\sigma(t, t)} - \rho(t + \tau', t) \frac{\sigma(t + \tau, t)}{\sigma(t, t)} \right.$$
$$\left. + 3\rho(t, t) \frac{\sigma(t + \tau, t)\sigma(t + \tau', t)}{2\sigma(t, t)^2} - \frac{\rho(t, t)}{2\sigma(t, t)} \left[ \sigma(t + \tau, t + \tau') - \sigma_K(\tau, \tau') \right] \right\} = 0, \quad (4)$$

where $\rho \equiv \sigma_\pi - \sigma_0$ (for large times). Here,

$$\sigma_K(t, t') = \begin{cases} \sigma(t, t') & \text{if } \theta > 1 - H \\ \sigma_s(t, t') & \text{if } \theta < 1 - H \end{cases} \quad (5)$$

Next, we find that the linear equation (4) admits solutions of the scaling form $\rho(t, t') = t^{2H - \theta} z_\theta(t/t')$, where $z_\theta(u)$ satisfies a linear integral equation of the form

$$\int_0^1 K_\theta(u, v) \left[ z_\theta(u) - z_\theta(1) \left( 1 - \frac{(2H - \theta)(1 - u)}{2} \right) \right] du = f_\theta(v), \quad (6)$$

where $K_\theta$ and $f_\theta$ are given in SI (Section D) in terms of $\sigma$. It is found that generic solutions $z_\theta(u)$ display divergences for small $u$, and we argue that $\theta$ is obtained by imposing that $z_\theta(u)$ is regular. We expect that this selection criterium is valid at least for $2H - \theta > 0$ since it amounts in this case to impose that $\rho(t, 0) = 0$. Self-consistency reasons also lead us to restrict the analysis to $H > 1/3$ (see SI, Section E). In practice, the linear integral equation Eq. (6) is solved numerically for a test value $\theta_{\text{test}}$ and yields a diverging solution $z_{\theta_{\text{test}}}(u) \sim A(\theta_{\text{test}}) u^{-\alpha(\theta_{\text{test}})}$; the persistence exponent $\theta$ is then obtained iteratively by enforcing that the prefactor vanishes, $A = 0$ (see SI, Section D). This finally provides a constructive, non perturbative determination of the persistence exponent $\theta$ for

Gaussian process with general non stationary dynamics, which is the central result of this paper.

## Applications

We now show how these results enable us to determine the impact of initial conditions in two physically relevant cases. The first type of problems (called type I here) is the determination of $\theta$ in systems which relax after a sharp temperature quench that occurs at initial time, which is a very generic situation that is in particular often realized to probe the aging dynamics of glassy systems[48]. Typically, physical realizations of the random process $x(t)$ can be the position of a monomer in various models of macromolecules or the local height of an interface, which span a number of values of $H$. In all these models, assuming that the initial state for $t \leq 0$ is an equilibrium state at temperature $T \neq 1$, while the dynamics at $t > 0$ occurs at temperature $T_0 = 1$, the covariance function for $t, t' > 0$ takes the form (see ref. 28 and SI, Section C)

$$\sigma(t,t') \propto T(t^{2H} + t'^{2H}) + (1-T)(t+t')^{2H} - |t-t'|^{2H}. \quad (7)$$

Of note, the temperature $T$ before the quench can be lower or larger than the temperature $T_0 = 1$ of the dynamics for $t > 0$. Examples of survival probabilities obtained from simulations are displayed in Fig. 1a, which clearly shows that the persistence exponent depends on the choice of initial conditions, and that the dependence of the persistence exponent on temperature is correctly predicted by our approach [Fig. 1b]. Remarkably, the values of $\theta$ for different temperatures span a large set of values and are markedly different from their value $\theta = 1 - H$ in the stationary state. In one example of simulations, we also recorded the trajectories in the future of the first-passage and measured numerically the function $z_\theta(x)$, which shows good agreement with the theoretical prediction [see Fig. 1c]. This figure also illustrates our procedure to determine $\theta$ as defined above: the calculated $z_\theta(u)$ show divergence for small $u$ whenever $\theta$ is above or below its exact value. In Fig. 1d, we check that our theory is also correct

for different $H$ (focusing on $T = 0$), for both superdiffusive and subdiffusive processes, and even far from the Markovian regime $H = 1/2$. In addition, explicit results can be obtained by analyzing our formalism [Eq. (4)] perturbatively in the limit $\varepsilon = H - 1/2 \to 0$. An expansion up to second order leads for any temperature $T$ before the quench to

$$\begin{aligned}\theta_1 = &1 - H - 2(\sqrt{2}-1)(1-T)(H-1/2) \\ &+ \{a_1(1-T)[a_2 + (1-T)]\}(H-1/2)^2 + \mathcal{O}\left((H-1/2)^3\right),\end{aligned} \quad (8)$$

where analytical expressions of $a_1$ and $a_2$ are given in SI (Section F), with numerical estimates $a_1 \simeq 1.77$, $a_2 \simeq 1.28$. Interestingly, in the particular cases $T = 0$ and $T = 1$, the first order terms coincide with the exact first order solution of ref. 28, which points towards the exactness of our approach at this order. These perturbative results are in good agreement with simulation results [Fig. 1b, d]. Finally, these results show that an imposed initial perturbation – here a temperature quench, deeply impacts the first-passage statistics of the system. In the case of subdiffusive (or antipersistent) dynamics ($H < 1/2$, realized typically in polymer models), it is found that, because of long range memory effects, an initial quench from a high ($T > 1$) to a low ($T_0 = 1$) temperature can strongly slow down the first-passage kinetics ($\theta < 1 - H$), while a quench from low to high temperatures accelerates the kinetics ($\theta > 1 - H$); opposite conclusions are reached for superdiffusive (or persistent) dynamics ($H > 1/2$).

The second class of problems (type II) corresponds to the determination of $\theta$ in an idealized situation where a given trajectory for $x(t < 0)$ is assumed to be accessible and observed at all times $t < 0$; this can be realized in various settings, ranging from single particle tracking techniques in the context of transport in complex systems, to the monitoring of the value of an asset in the context of financial markets. Here we aim at quantifying the impact of such observation of the system in the past ($t < 0$) on its future dynamics ($t > 0$). It is known[49–52]

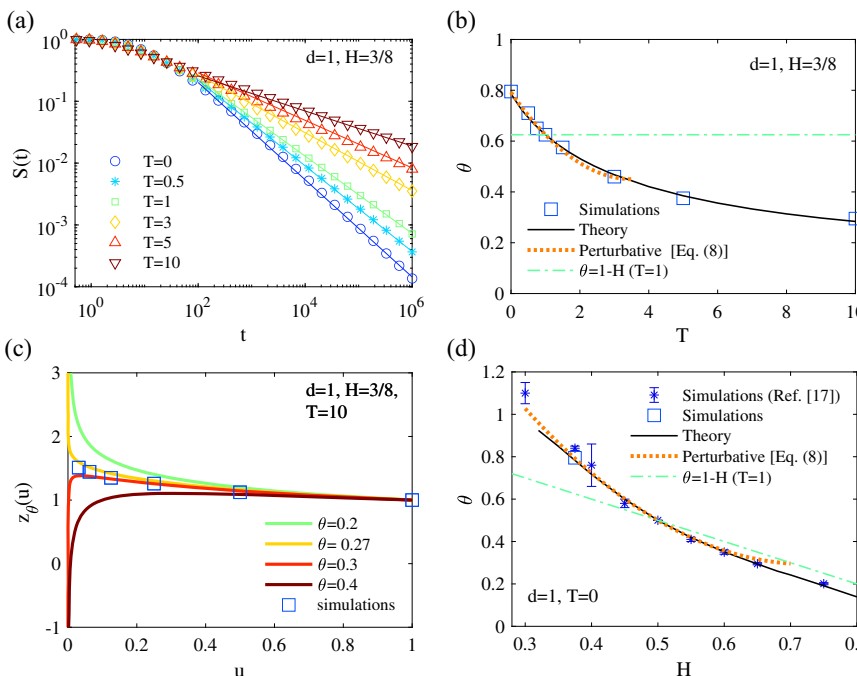

**Fig. 1 | Persistence for quenched fBM (type I problem) in $d = 1$. a** Example of survival probabilities for interface dynamics with $H = 3/8$ at different temperatures. The slopes of the continuous lines is the value of $\theta$ predicted in our approach. **b** Systematic comparison of $\theta$ as measured in simulations versus theoretical values for different $T$ and $H = 3/8$. **c** Value of $z_\theta(u)$ as measured in simulations (squares) by analyzing the statistics of trajectories after the FPT, compared with theoretical values for different $\theta$. Notice the divergences for small $u$ towards $\pm \infty$, which enable us to select the value of $\theta$ to minimize these divergences. **d** Same as **b** for different $H$, with $T = 0$ fixed. "Star" symbols are the simulation results of ref. 28 (the error bars indicate the confidence intervals given in table 1 of this reference).

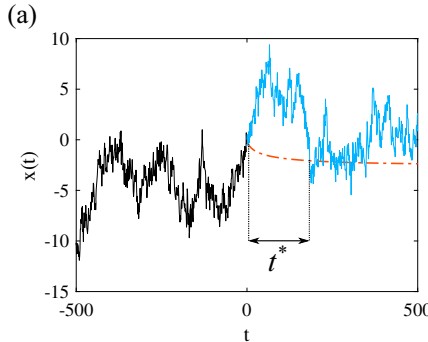

(a)

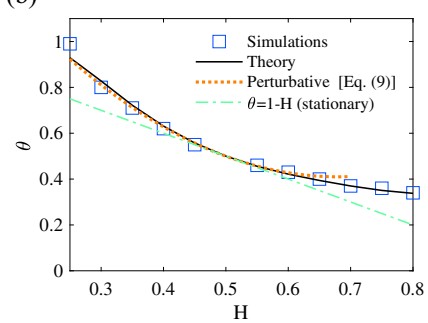

(b)

**Fig. 2 | Persistence for fBM conditioned on the past trajectory (type II problem).** **a** Definition of the problem: assume that a particular trajectory is observed for $t < 0$ (black curve). The blue line is one realization of trajectory for $t > 0$, the dashed red line represents the average trajectory given that the past trajectory is observed. The time $t^*$ is the first crossing time to this predicted average trajectory, and the persistence exponent characterizes the probability that $t^* > t$, for large $t$. **b** Comparison between values of $\theta$ for the fBM conditioned on the past trajectory, obtained in simulations (symbols), our theoretical approach (black line) and perturbation expansion (dashed red line). We also indicate the value $\theta = 1 - H$ for non-conditioned fBM.

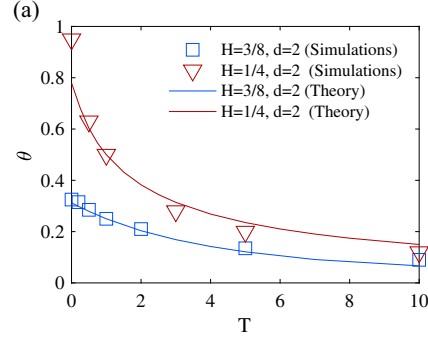

(a)

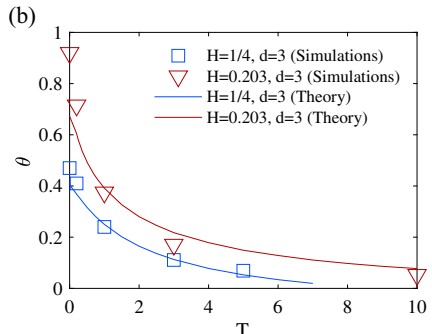

(b)

**Fig. 3 | Persistence exponents in dimensions higher than one.** **a** $d = 2$ and **b** $d = 3$. These results are obtained when the random walker is a tagged monomer in different polymer models: semi-flexible chains ($H = 3/8$), flexible chain without hydrodynamic interactions ($H = 1/4$), macromolecule of fractal architecture (Vicsek fractal, $H \simeq 0.203$).

that the mean future trajectory (for $t > 0$), conditional to a given observation in the past $x(t < 0)$, can be expressed as a linear combination of all positions in the past. Our approach makes it possible to determine quantitatively the exponent $\theta$ characterizing the probability $S(t)$ of not crossing this average conditional trajectory, or of not reaching a fixed threshold above (or below) it, see Fig. 2a. Strikingly, we find that the value of $\theta$ can be significantly larger than the value $\theta = 1 - H$ obtained in absence of any prior observation of the system. It does not depend on the particular realization of the observed past trajectory, but only on the fact that this observation is available. This thus shows that the very observation of the system can drastically impact the future first-passage statistics, and in fact effectively accelerate the dynamics at large times because $\theta \geq 1 - H$ for all values of $H$, irrespective of the persistent or antipersistent nature of the process. The results in Fig. 2b show again a good agreement between the predicted values of $\theta$ and simulations. As above a perturbation expansion of our formalism can be performed for weakly non-Markovian processes, leading to the explicit result

$$\theta_{II} = 1 - H + 4 \ln 2 (H - 1/2)^2 + \mathcal{O}((H - 1/2)^3), \tag{9}$$

which is supported by our simulations (Fig. 2b).

## Persistence in higher dimensions
Our theory can be generalized to the case of an isotropic Gaussian random process $\mathbf{x}(t)$ evolving in a space of dimension $d > 1$. In this case, to define the survival probability we replace the condition of reaching a threshold by the condition of reaching a target. To the best of our knowledge, in this case the persistence exponent has not been investigated in the literature for non-Markovian walks with non-stationary initial conditions, despite its obvious relevance to reactivity problems in complex systems. Here we restrict ourselves to the case where a target, even point-like, is found with probability one (compact case, when $dH < 1$). It turns out that very few changes are needed to generalize the theory in $d$ dimensions, generalized versions of the equations are presented in SI (Section C), and we restrict ourselves to $H > 1/(2 + d)$. Figure 3 shows simulation results when $\mathbf{x}(t)$ is the position of a monomer in various polymer models: semi-flexible or flexible chains, or fractal hyperbranched flexible macromolecules. It is found that our theory captures quantitatively the dependence of the persistence exponents on the temperature quench for all these models. This dependence on the temperature quench shows that the exponents describing the kinetics of absorption to a target are significantly modified by preparing the system with non-stationary initial conditions. The modification of persistence exponents with initial conditions could be relevant for the reactivity of complex macromolecules displaying widely distributed relaxation times such as proteins[53–55], in this context, non-equilibrium conditions could be obtained by a temperature quench, or by imposing a constraint, such as a geometric confinement or an external field, that is relaxed at $t = 0$. Our determination of the persistence exponent then allows to quantify the kinetics of reactions involving such molecules in such non-equilibrium conditions. Alternatively, if memory effects of the random walker come from its interactions with a surrounding viscoelastic medium, a non-

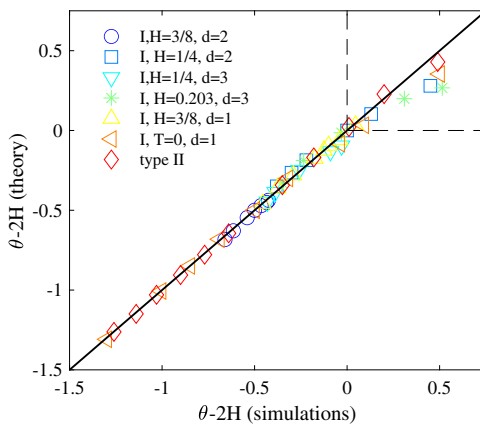

**Fig. 4 | Comparison of theoretical values of the persistence exponent versus simulations.** For all data shown in Figs. 1, 2, 3.

equilibrium initial state could be obtained by imposing a sharp change of the parameters characterizing this medium. In these cases, we predict that the reaction kinetics to a target can be deeply impacted, and display non-trivial exponents quantified by our approach (as soon as the covariance $\sigma$ can be calculated).

## Discussion

In Fig. 4, simulation data for $\theta$ for all the models considered in this work are recapitulated and compared with the values predicted by our approach. The data collapse shows an excellent agreement and validates our method. The slight departures of simulations from theory occur only when $2H - \theta > 0$ (see Fig. 4), in agreement with our previous remark that our selection criterium may not be valid anymore in this regime. Similarly, the curves which are the least precise on Fig. 3 are those for which $H$ is very close to $1/(2 + d)$, where the theory is not expected to give accurate results anymore. Altogether, this shows that our theory provides a non perturbative, constructive, quantitative determination of the persistence exponents for general Gaussian stochastic processes with non-stationary initial conditions, which typically model the relaxation after an initial perturbation of systems with non Makovian dynamics, such as tracer particles in complex environments with many interacting degrees of freedom. It would be interesting to determine if our approach could be applied to other processes displaying transient, long-lived aging properties such as those observed in glasses after quenching protocols[48]. Our results demonstrate that initial perturbations can have a deep, long lived impact on the first-passage statistics of non-Markovian processes. Importantly, our theory also predicts non-trivial exponents in dimension higher than one, and thus opens the way to the quantification and control of reaction kinetics for complex systems with non-equilibrium initial conditions.

## Methods

### Numerical measurement of persistence exponents

In order to measure the persistent exponent for type I processes, we have performed stochastic simulations of (i) the Edwards-Wilkinson interface, or equivalently a flexible polymer chain of beads and springs without hydrodynamic interactions ($H = 1/2$), (ii) the Mullins-Herring dynamics ($H = 3/8$), (iii) a macromolecule represented as a bead spring network whose connectivity is the same as that of a Vicsek fractal of functionality $f = 4$ for which we used the method described in ref. 14 to generate the stochastic trajectories. These stochastic processes, as well as the simulation algorithms are described in SI, Section C.

### Theoretical estimate of $\theta$ and perturbative analysis of the theory

To evaluate numerically the persistence exponent $\theta$ from Eq. (6), we have proceeded as follows: for a given test value $\theta_{\text{test}}$ we solve the

integral equation numerically, this generally yields a solution that diverges for small $u$: $z_{\theta_{\text{test}}}(u) \sim A(\theta_{\text{test}}) u^{-\alpha(\theta_{\text{test}})}$. Then the persistence exponent is selected iteratively by choosing the value of $\theta$ so that $A = 0$ (see SI for details, Section D).

### Perturbative analysis of the formalism

Explicit analytical results were obtained in the limit $\varepsilon = H - 1/2 \to 0$ by inserting into Eq. (4) the ansatzs $\sigma_s(t,t') \simeq \min(t,t') + \sum_{n\geq 1} \varepsilon^n \sigma_{s,n}(t,t')$, $\sigma(t,t') = \sigma_s(t,t') + \sum_{n\geq 1} \varepsilon^n \omega_n(t,t')$, $\theta = 1 - H + \sum_{n\geq 1} \varepsilon^n \delta_n$ and $\rho(u,v) = \sum_{n\geq 0} \varepsilon^n \rho_n(u,v)$. The resulting equations for $\rho_0$ and $\rho_1$ can be solved analytically and, as in the non-perturbative approach, the value of the persistence exponent is chosen to ensure that $\rho(u,v)$ does not diverge in the limit $u \to 0$, leading to Eqs. (8) and (9). The calculation details are provided in SI, Section F.

## Data availability

The data used to measure numerically the values of the persistence exponents shown in Figs. 1–4 [https://doi.org/10.5281/zenodo. 6761006, link: https://zenodo.org/record/6761006] have been deposited in the zenodo database [https://doi.org/10.5281/zenodo. 6761006, link: https://zenodo.org/record/6761006].

## Code availability

The codes used to measure numerically the values of the persistence exponents shown in Figs. 1–4, and a program that solves numerically Eq. (6) have been deposited in the zenodo database [https://doi.org/ 10.5281/zenodo.6761006, link: https://zenodo.org/record/6761006].

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

## Acknowledgements

T-V. M. and T. G. acknowledge the support of the grant *Complex-Encounters*, ANR-21-CE30-0020-01. Computer time for this study was provided by the computing facilities MCIA (Mesocentre de Calcul Intensif Aquitain) of the Université de Bordeaux and of the Université de Pau et des Pays de l'Adour.

## Author contributions

All authors contributed to analytical calculations. N.L., T.M., and T.G. performed numerical computations. O.B., R.V., and T.G. conceived research and wrote the manuscript.

## Competing interests

The authors declare no competing interests.
