## [Peer Review File · Nature Communications]

REVIEWER COMMENTS

Reviewer #1 (Remarks to the Author):

In this paper, the authors develop a theory to study the long term behavior of stochastic processes which are Gaussian, non-Markovian, and in d -dimensional space. In particular, the authors are interested in calculating so-called persistence exponents, which are the exponents on the algebraic decay of the probability that the stochastic process has not reached a target before a large time (sometimes called the survival probability of the persistence of the stochastic process). The authors then employ their theory to demonstrate that initial perturbations can strongly affect this persistence exponent.

I think that this is a very interesting paper and the authors employ a variety of novel ideas. I actually like this paper very much. However, I do not think the authors make a strong case that this paper has broad enough appeal to warrant publication in this journal. While the results and methods are certainly interesting, I am not sure the somewhat heuristic calculation of the persistence exponent of a class of stochastic processes has a very broad appeal. The results seem better suited for a more specialized journal. In addition, I think that the exposition could be improved as I found some parts of the paper difficult to read. Below, I list some more minor points.

-Equation (2) is the starting point in the main text, which involves the probability density that the stochastic process is at 0 at time t . This equation also involves the FPT density F , but as far as I can tell, it has not yet been stated what the FPT is the FPT "to". That is, the reader has not yet been told that it is the FPT to 0.

-This is very minor, but F denotes the FPT density in the main text, whereas f denotes the FPT density in the Supplementary Information.

Reviewer #2 (Remarks to the Author):

The authors consider the persistence of non-Markovian Gaussian stochastic processes after initial perturbations like a temperature quench or the observation of the past trajectory. They find analytically and numerically that the persistence exponent θ varies continuously with the temperature of the quench and differs substantially from its value in the stationary (non-quenched) case, which it also does after the observation of the past trajectory. Since θ determines the first passage statistics it is thus shown that the considered initial perturbations have a deep impact on them, too. Moreover, the authors are able to generalize their results from one dimensional processes to higher dimensions $d > 1$. Since the considered processes can represent the position of a monomer in various polymer models the dependence of θ on the temperature quench shows that kinetics of absorption to a target is significantly modified when the system is prepared in non-stationary initial conditions.

The paper is well written and accessible, with some effort, also to non-experts, in particular because all calculations are deferred to the supplementary material / information. The main novelty of the paper is the method with which the persistence exponents are calculated, which is only sketched in the main text, but elaborated in full detail in the SI, which I find it a bit risky since these extensive and technically challenging calculations are then neither proofread nor edited by the journal. But as said: it certainly improves readability.

The mentioned analytical calculation is a true tour de force comprising a number of elegant hypotheses (like the process in the future of the FTP to be Gaussian and non-standard

solution procedures (like adjusting theta such that z in 6 is regular) – which are nevertheless tested numerically. The resulting agreement of the resulting predictions with the numerical data is excellent and convincing.

It is known from decade-long research on aging in glasses and spin glasses (which is not mentioned in the manuscript) that quenches in temperature, field or other parameters have a deep and everlasting impact on the dynamics (which stays out of equilibrium forever), but a quantification in terms of persistence or first passage times, has not, to my knowledge, been performed in this context. Therefore, the results presented in this paper are groundbreaking and will influence future research also in other areas. I recommend its publication in Nat. Comm.

Minor comments:

- **For the sake of generality the introduction refers always to “processes” but on p.3, 1st sentence the first time a “random walker” is mentioned without further specification. This is somewhat inconsistent.**
- **Eq(2): F is the first passage time density – which first passage time (i.e. which event is referred to)?**
- **Eq(7) is crucial, since here the temperature dependence of the covariance appears the first time (from which all following temperature dependencies emerge). Although derived in [17] it would be good to spend a few words on the physical origin of the temperature dependence, for instance in the context of the mentioned polymer or interface models. As it stands it just looks like a formal parameter entering the covariance. Note that section C of the SI does not provide an explanation either.**

Answer to reviewer's comments

Reviewer #1 (Remarks to the Author):

In this paper, the authors develop a theory to study the long term behavior of stochastic processes which are Gaussian, non-Markovian, and in d -dimensional space. In particular, the authors are interested in calculating so-called persistence exponents, which are the exponents on the algebraic decay of the probability that the stochastic process has not reached a target before a large time (sometimes called the survival probability of the persistence of the stochastic process). The authors then employ their theory to demonstrate that initial perturbations can strongly affect this persistence exponent.

I think that this is a very interesting paper and the authors employ a variety of novel ideas. I actually like this paper very much.

We thank the reviewer for these positive comments.

However, I do not think the authors make a strong case that this paper has broad enough appeal to warrant publication in this journal. While the results and methods are certainly interesting, I am not sure the somewhat heuristic calculation of the persistence exponent of a class of stochastic processes has a very broad appeal. The results seem better suited for a more specialized journal. In addition, I think that the exposition could be improved as I found some parts of the paper difficult to read.

We respectfully disagree with this statement that our paper does not have « a broad enough appeal to warrant publication in this journal » for the following reasons:

1. The problem of persistence of stochastic processes is clearly recognized as a major problem of statistical mechanics (Refs 1,2,8,9,...), with a very broad range of applications in different fields, from coarsening dynamics to fluctuating interfaces. It is recognized that persistence exponents provide a key, non-trivial quantification of the dynamics of such systems. Determining persistence exponents for general non-Markovian dynamics — which is the rule rather than the exception — is an outstanding theoretical problem. In this paper, we go one step further and provide a new methodology to determine the persistence exponent of non stationary processes — more precisely processes that relax to a stationary states after an initial perturbation. This situation corresponds physically to the generic case of temperature quenches, which is a typical scenario involved in a broad range of experimental contexts. In addition to the technically challenging aspects of our work, this clearly pleads for its broad appeal. This opinion is shared by the other referee, who states that our results are « groundbreaking and will influence future research also in other areas ».
2. Furthermore, we also provide persistent exponents *in space dimensions higher than one*, an aspect which was so far completely unexplored in the field of persistence exponents. This enables us to connect the field of persistence exponents and the field of reactivity of complex polymers and macromolecules, which are known to display large relaxation times. Our work thus significantly broadens the field of persistence to the relevant $d=2,3$ space dimensions, which opens the way to new experimental applications.
3. Finally, we would like to clarify the status of our results, which the referee dubs « heuristic ». Our methods rely on the central hypothesis that the process in the future of the first passage is Gaussian. This hypothesis is confirmed numerically and thus well controlled. Additionally, our results are fully consistent with exact perturbative expressions obtained for weakly non

Markovian processes. Last, we add that our results are very general and applicable to Gaussian processes, which are paradigmatic models in the field of persistence. This strengthens the broad impact of our work.

To clarify these points, in the revised version, we (1) highlight the broad importance of persistence exponents in various fields (see page 2) (2) emphasize new applications of persistence exponents in dimensions 2,3 such as reaction kinetics of macromolecules (see page 8), (3) mention possible applications in the context of glasses (page 9).

Below, I list some more minor points.

-Equation (2) is the starting point in the main text, which involves the probability density that the stochastic process is at 0 at time t . This equation also involves the FPT density F , but as far as I can tell, it has not yet been stated what the FPT is the FPT "to". That is, the reader has not yet been told that it is the FPT to 0.

-This is very minor, but F denotes the FPT density in the main text, whereas f denotes the FPT density in the Supplementary Information.

Thank you for noting these points. They have been corrected.

Reviewer #2 (Remarks to the Author):

The authors consider the persistence of non-Markovian Gaussian stochastic processes after initial perturbations like a temperature quench or the observation of the past trajectory. They find analytically and numerically that the persistence exponent θ varies continuously with the temperature of the quench and differs substantially from its value in the stationary (non-quenched) case, which it also does after the observation of the past trajectory. Since θ determines the first passage statistics it is thus shown that the considered initial perturbations have a deep impact on them, too. Moreover, the authors are able to generalize their results from one dimensional processes to higher dimensions $d > 1$.

Since the considered processes can represent the position of a monomer in various polymer models the dependence of θ on the temperature quench shows that kinetics of absorption to a target is significantly modified when the system is prepared in non-stationary initial conditions.

The paper is well written and accessible, with some effort, also to non-experts, in particular because all calculations are deferred to the supplementary material / information. The main novelty of the paper is the method with which the persistence exponents are calculated, which is only sketched in the main text, but elaborated in full detail in the SI, which I find it a bit risky since these extensive and technically challenging calculations are then neither proofread nor edited by the journal. But as said: it certainly improves readability.

The mentioned analytical calculation is a true tour de force comprising a number of elegant hypotheses (like the process in the future of the FTP to be Gaussian and non-standard solution procedures (like adjusting θ such that z in 6 is regular) – which are nevertheless tested numerically. The resulting agreement of the resulting predictions with the numerical data is excellent and convincing.

It is known from decade-long research on aging in glasses and spin glasses (which is not mentioned in the manuscript) that quenches in temperature, field or other parameters have a deep and

everlasting impact on the dynamics (which stays out of equilibrium forever), but a quantification in terms of persistence or first passage times, has not, to my knowledge, been performed in this context. Therefore, the results presented in this paper are groundbreaking and will influence future research also in other areas. I recommend its publication in Nat. Comm.

We thank the referee for his careful reading of the manuscript and the positive comments. We have now added a sentence stating the possible opening of our approach to glasses in the conclusion.

Minor comments:

- *For the sake of generality the introduction refers always to “processes” but on p.3, 1st sentence the first time a “random walker” is mentioned without further specification. This is somewhat inconsistent.*
- *Eq(2): F is the first passage time density – which first passage time (i.e. which event is referred to)?*

Thanks for these remarks. These points have been corrected.

- *Eq(7) is crucial, since here the temperature dependence of the covariance appears the first time (from which all following temperature dependencies emerge). Although derived in [17] it would be good to spend a few words on the physical origin of the temperature dependence, for instance in the context of the mentioned polymer or interface models. As it stands it just looks like a formal parameter entering the covariance. Note that section C of the SI does not provide an explanation either.*

The new version now provides details on how to derive the covariance appearing in Eq. (7) in SI.

REVIEWERS' COMMENTS

Reviewer #1 (Remarks to the Author):

The authors have addressed my comments. I recommend the paper for publication.

Reviewer #2 (Remarks to the Author):

The authors have fully complied with my suggestions contained in my previous report. For the reasons I already elaborated therein I recommend the publication of the manuscript in Nature Communications.

Response to referees

REVIEWERS' COMMENTS

Reviewer #1 (Remarks to the Author):

The authors have addressed my comments. I recommend the paper for publication.

Our answer: We thank the referee for this recommendation.

Reviewer #2 (Remarks to the Author):

The authors have fully complied with my suggestions contained in my previous report. For the reasons I already elaborated therein I recommend the publication of the manuscript in Nature Communications.

Our answer: We thank the referee for this recommendation.